# AUTOMATED STRUCTURAL LEARNING OF STOCHASTIC DYNAMICS

**Xeniya Bashkova & Alexander Hvatov**
NSS Lab, ITMO University
St. Petersburg, Russian Federation
`alex_hvatov@itmo.ru`

## ABSTRACT

While automated equation discovery has successfully extracted deterministic models from time-series data, extending these methods to stochastic systems remains a critical problem. Existing symbolic regression tools typically yield rigid Ordinary Differential Equations (ODEs) or disjointed ensembles, failing to capture intrinsic system uncertainty without sacrificing analytical interpretability. To resolve this, we introduce a framework that recasts equation discovery as a probabilistic inference task. This approach harnesses the inherent stochasticity of evolutionary optimization, mapping the structural variability across multiple symbolic regression executions into continuous probability distributions over mathematical terms and coefficients. This mechanism bypasses restrictive parametric assumptions, directly distilling noisy observations into a single, cohesive Stochastic Differential Equation (SDE). Unlike opaque neural architectures or unwieldy ODE ensembles, our framework generates compact, transparent models that are natively equipped for formal symbolic manipulation and rigorous mathematical reasoning.

## 1 INTRODUCTION

Time-series forecasting is a cornerstone of quantitative research across multiple scientific domains. Yet, a fundamental challenge persists: real-world systems are intrinsically stochastic. The data we observe are merely single realizations of complex processes where underlying deterministic laws are continuously perturbed by random fluctuations. Traditional data-driven approaches—ranging from classical statistics like ARIMA to deep learning architectures like ANNs and LSTMs—structurally treat these observations as deterministic signals to be interpolated. By mathematically optimizing strictly for the "mean" behavior, they systematically blind themselves to the full spectrum of underlying uncertainty Suryadevara & Yanamala (2020); Sherstinsky (2020); Yadav & Thakkar (2024), a flaw that becomes critical when modeling chaotic or highly non-linear dynamics.

The necessity to capture this uncertainty has driven the adoption of state-space models (SSMs), which excel at inferring hidden states in noisy environments Hu et al. (2024). However, their reliance on expert-driven parameterization often creates an analytical bottleneck Gu & Dao (2023). Broadly, modern stochastic machine learning branches into ensemble methods, diffusion models, and approaches based on Stochastic Differential Equations (SDEs). While ensembles and diffusions have set new benchmarks in uncertainty quantification—as evidenced in recent ProbTS evaluations Lin et al. (2024); Zhang et al. (2024)—SDEs offer something uniquely valuable: a rigorous, continuous-time mathematical framework.

SDEs remain the gold standard for modeling noise-perturbed systems, with widespread applications in finance Yau et al. (2024); Buccheri et al. (2024), physics Haider et al. (2024), engineering Iversen et al. (2017), and biology Jha & Langmead (2012). They are remarkably versatile, capable of integrating background Markov processes Henshaw et al. (2024), sentiment shifts Kalaycı et al. (2024), external spatial shocks Cerqueti et al. (2024), and reliability decay in masked multi-component systems Yang et al. (2024). Despite this immense analytical power, their use is heavily confined to specific domains where the governing structural equations are already theoretically known, severely limiting their general applicability. The fundamental challenge lies in extracting these explicit governing stochastic structures directly from raw observational data.

To bridge the gap between flexible black-box algorithms and rigid, theoretical SDEs, hybrid architectures have emerged. Neural SDEs and VAE-based models parameterize drift and diffusion functions using deep neural networks, successfully capturing complex stochastic behaviors such as those observed in financial ETFs Sun et al. (2024); Gierjatowicz et al. (2020); Liu et al. (2019); Hasan et al. (2020). However, this flexibility comes at a steep cost: the complete loss of mathematical interpretability. These models fail to yield an explicit, human-readable structural form, rendering them unsuitable for formal analytical reasoning.

A parallel breakthrough has occurred in the field of automated Equation Discovery. Algorithms like SINDy Brunton et al. (2016) and EPDE Maslyaev et al. (2021) excel at structural learning for deterministic systems, distilling raw data into elegant, interpretable Ordinary Differential Equations (ODEs). To improve robustness against noise, ensemble extensions like E-SINDy Fasel et al. (2022) generate massive collections of candidate equations. While statistically sound, these multitudes of models are utterly impractical for symbolic manipulation. They essentially mask structural uncertainty behind an unwieldy empirical ensemble. The leap from learning deterministic structural forms (or generating large ODE ensembles) to identifying a single, fully stochastic, and analytically tractable differential equation remains an open problem.

This reveals a critical methodological gap: the scientific community lacks an automated structural learning framework capable of discovering compact, interpretable SDE systems directly from time-series data. Current solutions force a compromise—they either sacrifice structural transparency (Neural SDEs), ignore intrinsic noise (deterministic symbolic regression), or produce outputs too convoluted for formal mathematical reasoning (ensembles).

In this paper, we address this gap by proposing a novel stochastic structural learning framework. Our approach moves beyond black-box approximations and cumbersome deterministic ensembles to autonomously discover a single, mathematically rigorous SDE that captures the true stochastic essence of the underlying system. By reframing equation discovery as a probabilistic structural inference problem, this framework represents a paradigm shift: it seamlessly fuses the analytical transparency of classic mathematical modeling with the scalability of modern data-driven generative discovery, explicitly extracting the mathematical structure of both deterministic trends and stochastic fluctuations.

## 2 THEORY

The primary goal of equation discovery is to recover a differential equation that governs the evolution of a dynamical system based solely on observed data. Given a time series $\{x(t_i)\}_{i=0}^{N}$, the objective is to find a symbolic equation of the form:

$$\mathcal{L}[x](t) = \sum_{i=1}^{M} p_i(t) \, d_i[x](t) = f(t), \tag{1}$$

Here, $d_i[x](t)$ represent the basic mathematical building blocks (e.g., $d_i[x] \in \{x, \dot{x}, x\dot{x}, \ddot{x}, \ldots\}$) selected from a predefined library of candidate terms. The coefficients $p_i(t)$ are explicit time-dependent functions, such as polynomials or harmonic functions. These coefficients exist within the span of a foundational basis:

$$p_i \in \mathrm{span}(\{1, t, t^2, \sin \omega t, \ldots\}) \tag{2}$$

This deterministic formulation outputs a single best-fit model for the observed data. Like any standard machine learning model, it represents a combination of the optimal structural features $\mathcal{T}[x](t)$ and the residual noise, captured by $\mathcal{N}[x](t)$. This noise originates from various sources, including term library incompleteness, inherent data noise, and numerical differentiation errors:

$$\mathcal{L}[x](t) = \mathcal{T}[x](t) + \mathcal{N}[x](t) \tag{3}$$

To isolate the noise component, the equation discovery process is often repeated multiple times using different data subsampling techniques, such as bootstrapping or noise injection. This iterative

discovery yields an ensemble of deterministic equations, a concept utilized in methods like E-SINDy Fasel et al. (2022), represented as:

$$E_m = \{\mathcal{L}_1[x](t), \dots, \mathcal{L}_K[x](t)\}, \tag{4}$$

which subsequently produces a corresponding ensemble of solved trajectories:

$$E_f = \{\bar{x}_1(t), \dots, \bar{x}_K(t)\}. \tag{5}$$

However, we note that prior ensemble-based approaches often overlook the distinct nature of $\mathcal{T}[x](t)$ and $\mathcal{N}[x](t)$. Consequently, the intrinsic statistical properties of the equation ensemble $E_m$ equation 4 are largely ignored. For instance, in E-SINDy, equations are frequently truncated and forced into a predetermined structural template (such as the Lotka-Volterra equations) to accommodate specific solvers. This imposes excessively strict prior assumptions on the structural form $\mathcal{T}[x](t)$ and risks discarding vital dynamic information by absorbing significant functional terms into the noise component $\mathcal{N}[x](t)$.

Fundamentally, if the learned structure is correct, the noise component $\mathcal{N}[x](t)$ should exhibit a zero mean. If it does not, it implies that uncaptured, significant deterministic terms remain, which must be structurally learned and explicitly moved into $\mathcal{T}[x](t)$.

Regardless of how the solution ensemble $E_f$ equation 4 is generated, it merely acts as a collection of samples from a stochastic process, characterized by an average trajectory $\mu(t)$ and variability $\sigma(t)$. Critically, this ensemble of solutions alone is insufficient to reconstruct the distinct symbolic components $\mathcal{T}[x](t)$ and $\mathcal{N}[x](t)$ defined in equation 3.

**Stochastic Structural Learning.** To overcome this limitation, we introduce a structural learning framework that directly embeds uncertainty into the discovered mathematical architecture. Instead of relying on deterministic, fixed coefficients $p_i(t)$, we elevate them to random variables or stochastic functions $P_i(t)$. The governing equation is structurally redefined as:

$$\mathcal{L}_{\mathcal{E}}[x](t) = \sum_{i=1}^{M} P_i(t)\, d_i[x](t) = f(t), \tag{6}$$

where $P_i(t)$ naturally accounts for the variability across different realizations of the system. The result is no longer a disjointed ensemble, but a single, unified symbolic structure with stochastic coefficients. We retain the physical building blocks $d_i[x](t)$, but the algorithmic variability previously spread across multiple models is now formally parameterized within the equation itself.

To formalize this structural learning phase, we define the coefficients as belonging to a *stochastic span*:

$$P_i(t) \in \mathrm{sspan}(\{1, t, t^2, \sin(\omega t), \dots\}), \tag{7}$$

The stochastic span is mathematically defined as:

$$\mathrm{sspan}(A) = \sum_{a_i \in A} \delta_i a_i, \;\; \delta_{ij} = \delta_i \delta_j \tag{8}$$

This implies that in the linear combination of basis vectors $a_i$ from space $A$, the standard scalar coefficients are replaced by independent random variables $\delta_i$. For practical implementation, we assume normally distributed variables $\delta_i \sim \mathcal{N}(\mu, \sigma)$.

While the representation in equation 6 is highly efficient for symbolic regression, it can be structurally decomposed into a form more native to Stochastic Differential Equation (SDE) theory. By separating the mean from the variance for each $\delta_i$:

$$\delta_i = \mu_i(t) + \sigma_i(t) \;, \sigma_i(t) \sim \mathcal{N}(0, \sigma_i(t)) \tag{9}$$

we isolate the deterministic trend from the stochastic fluctuations. This yields a structural decomposition of the learned operator:

$$\mathcal{L}_{\mathcal{E}}[x](t) = \mathcal{L}_\mu[x](t) + \mathcal{L}_\sigma[x](t) \tag{10}$$

Here, $\mathcal{L}_\mu[x](t)$ functions as a standard deterministic ODE equation 1, while the stochastic component $\mathcal{L}_\sigma[x](t)$ can be formulated as:

$$\mathcal{L}_\sigma[x](t) = \sum_{i=1}^{M} P_i(t)\, d_i[x](t)\sigma_i(t)dW_i, \tag{11}$$

The learned set of terms in equation 11 is preserved, but the structural variability is now explicitly modeled via separate Wiener processes $dW_i$. Consequently, our structural learning pipeline seamlessly bridges the compact symbolic form equation 6 to a rigorous Stratonovich SDE form equation 11. This structural equivalence provides several theoretical guarantees.

**Existence, Uniqueness, and Convergence.** We briefly address the theoretical consistency of this framework, demonstrating how the following diagram commutes:

$$
\begin{array}{ccc}
E_m & \xrightarrow{\texttt{ODEint}} & E_f \\
{\scriptstyle (?)K\to\infty}\Big\downarrow & & \Big\downarrow{\scriptstyle (?)K\to\infty} \\
\mathcal{L}_{\mathcal{E}}[x](t) & \xrightarrow{\texttt{SDEint}} & x_{\mathcal{E}}(t)
\end{array}
\tag{12}
$$

The primary analytical question is how to interpret the limit $E_m \xrightarrow{K\to\infty} \mathcal{L}_{\mathcal{E}}[x](t)$. This fundamentally asks: "Is the noise component $\mathcal{N}[x](t)$ within every ensemble equation $E_m$ small enough to be safely neglected?" Generally, the answer is no. A highly stochastic time series rarely corresponds to a single, unique ODE; it manifests across several "structural modes". Therefore, the structural learning process must first cluster the ensemble into stable structural forms (distinct sets of active terms) before parameterizing the stochastic models via equation 6. Once isolated into a stable structural mode, the resulting SDE models achieve uniqueness in the limiting case, driven by the Central Limit Theorem.

This clustering logic also clarifies the limit $E_f \xrightarrow{K\to\infty} x_{\mathcal{E}}(t)$. Because our learned structure converges to the Stratonovich form equation 11, the solution to the resulting SDE can be rigorously approximated using sampled ODEs—a direct consequence of the Wong-Zakai theorem Wong & Zakai (1965). The validity of this limit depends on whether the ensemble $E_f$ accurately represents samples drawn from $\mathcal{L}_{\mathcal{E}}[x](t)$.

Since the coefficients in equation 6 are derived from the ensemble $E_f$, the accuracy hinges on correctly modeling the underlying distributions. Provided we restrict the learning to a single, stable structural mode, the Central Limit Theorem justifies the use of continuous normal distributions for independent term appearances. If multiple structural modes exist simultaneously, conditional term dependencies arise; our framework addresses this by isolating these modes to ensure coefficient independence.

Finally, regarding the rate of structural convergence—i.e., the ensemble size $K$ necessary to stabilize the stochastic parameters—we must account for errors introduced by both coefficient estimation and numerical differentiation. Assuming a consistent data stochasticity model and fixed differentiation algorithms, we empirically established that an ensemble size of $K \sim 50$ provides an optimal trade-off between structural stability and computational feasibility.

Ultimately, this structural learning methodology reformulates the equation discovery challenge as a probabilistic inference task. By harnessing evolutionary stochasticity, it systematically explores the space of plausible mathematical structures while simultaneously quantifying epistemic uncertainty in the terms and establishing rigorous probabilistic bounds on the model's parameters.

## 3 REALIZATION

The proposed structural learning framework is explicitly designed for the analysis of stochastic dynamical systems, bridging the gap between automated equation discovery and rigorous uncertainty quantification. The operational pipeline begins with the discovery phase, where input time-series data is processed using symbolic regression algorithms such as EPDE or E-SINDy. Through sparse regression or evolutionary optimization, these tools generate an initial ensemble of candidate deterministic ODE structures equation 4.

Subsequently, statistical aggregation is applied to compute the mean coefficient values and their variances across the entire generated ensemble. To distill a robust set of governing mathematical forms, a threshold-based filtering mechanism eliminates equations exhibiting mathematically unstable coefficients or implausible parameter means. To further enhance robustness against local fluctuations, the time series can be segmented into discrete intervals during the equation search phase, synthesizing the final global equation over the entire temporal domain. The resulting deterministic ODEs, typically extracted via EPDE, act as the system's structural backbone. They establish a baseline dynamic trajectory that is isolated from stochastic interference, as depicted in Fig. 1a.

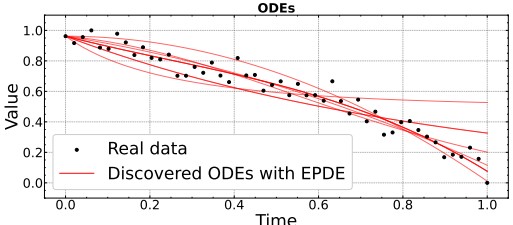

(a) Solution of the discovered deterministic ODE system.

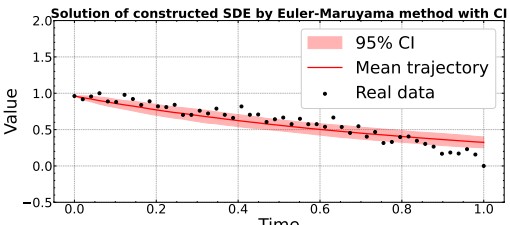

(b) Numerical solution of the derived SDE (Euler-Maruyama).

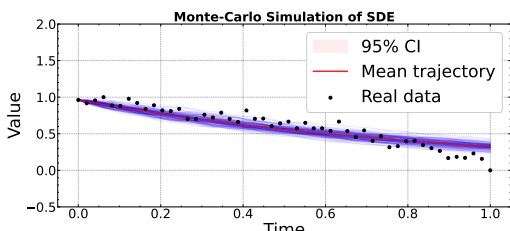

(c) Uncertainty quantification via Monte Carlo ensemble.

Figure 1: (a) Solution of the deterministic ODEs discovered by EPDE from the original data (black dots). (b) Single realization of the SDE solution obtained by the Euler-Maruyama method. (c) Statistical ensemble of solutions from the Monte Carlo simulation, providing a mean trajectory and uncertainty bounds.

Once this deterministic foundation is established, the framework transitions to modeling the full Stochastic Differential Equation. A core theoretical postulate here is that realistic macroscopic noise typically stems from rapid, unresolved microscopic dynamics continuously perturbing the system's underlying parameters. This physical intuition necessitates modeling the stochasticity in the Stratonovich interpretation, which mathematically adheres to the classical chain rule and is uniquely suited for systems driven by finite-correlation-time noise. Consequently, we initially formulate the model in the standard Itô form:

$$dX_t = a(X_t, t)dt + b(X_t, t)dW_t, \qquad (13)$$

where $X_t$ denotes the state variable, $a(\cdot)$ dictates the deterministic drift, $b(\cdot)$ defines the diffusion coefficient, and $W_t$ is a standard Wiener process. We then rigorously transform this into its Stratonovich equivalent:

$$dX_t = \tilde{a}(X_t, t)dt + \tilde{b}(X_t, t) \circ dW_t, \tag{14}$$

where the modified coefficients $\tilde{a}$ and $\tilde{b}$ now inherently encapsulate both the deterministic averages and the stochastic perturbations. This formal conversion is achieved by incorporating the Wong-Zakai correction into the drift term.

The derived Stratonovich SDE is subsequently solved using numerical integration. As illustrated in Fig. 1b, we employ the Euler-Maruyama method, a foundational scheme for SDE trajectory simulation. Finally, to ensure robust uncertainty quantification, the framework performs a large-scale Monte Carlo simulation, generating comprehensive probabilistic bounds for the predicted states.

A detailed architectural schematic of the complete operational pipeline is provided in Appendix Fig. A.

## 4 RESULTS

**E-SINDy Comparison.** To validate the proposed structural learning algorithm, we first evaluated it on the classical Lotka-Volterra predator-prey system. The resulting SDE system discovered by our framework is presented in Fig. 2, requiring an ensemble size of merely 35 sampled equations.

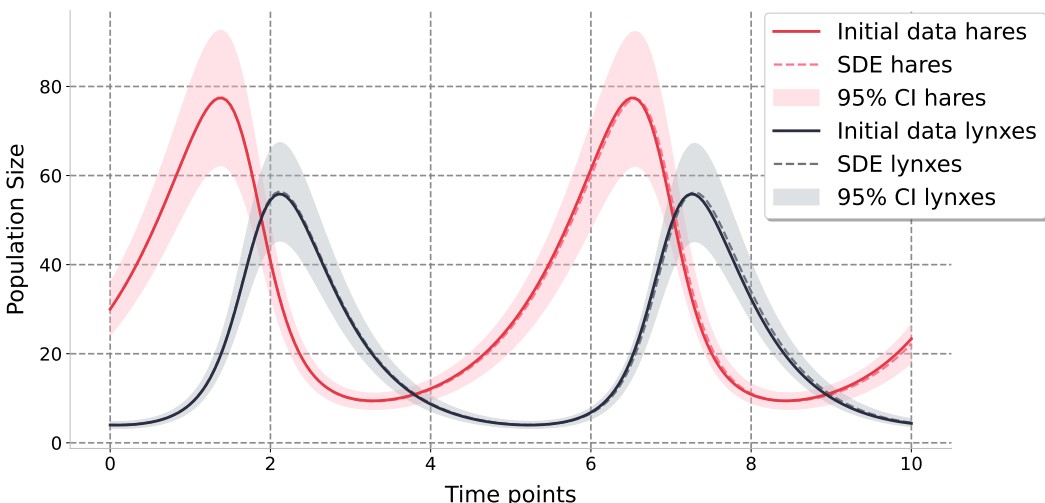

Figure 2: The resulting system of Lotka-Volterra equations obtained using the structural learning framework.

The explicitly discovered mathematical structure is defined in equation 15:

$$\begin{cases} \dot{u} = (0.5912 \pm 0.2642)\, u - (0.0314 \pm 0.1772)\, u\, v, \\ \dot{v} = -(0.8314 \pm 0.0913)\, v + (0.02814 \pm 0.1677)\, u\, v \end{cases} \tag{15}$$

Here, the coefficients are treated as normally distributed random variables, denoted as $\mu \pm 1.96\sigma^2$, where $\mu$ represents the mean parameter value and $1.96\sigma^2$ indicates the 95% confidence interval computed across the sampled ODE solutions.

We compared these results with those generated by E-SINDy Fasel et al. (2022), shown in equation 16. For this specific demonstration, E-SINDy was executed using 1000 library bagging runs without data bootstrapping.

$$\begin{cases} \dot{u} = (0.5274 \pm 0.0049)\, u - (0.025 \pm \varepsilon)\, v\, u, \\ \dot{v} = -(0.9691 \pm 0.1779)\, v + (0.027 \pm \varepsilon)\, v\, u \end{cases} \tag{16}$$

Using the identical notation framework for coefficient distributions, a Kolmogorov-Smirnov test confirmed that the parameter distributions synthesized by the two algorithms differ with statistical significance. As a benchmark, the theoretically ideal parameterization, derived from rigorous statistical analysis Howard (2009), is established in equation 17:

$$\begin{cases} \dot{u} = 0.55u - 0.028uv \\ \dot{v} = -0.84v + 0.026uv \end{cases} \tag{17}$$

A comparison of the mean parameter values across equation 15, equation 16, and equation 17 indicates that the coefficients discovered by our structural approach equation 15 align more closely with the theoretical ideal. While working with real-world data implies the absence of a singular "correct" ground truth—meaning both E-SINDy and our method technically identify valid governing SDEs—a crucial operational distinction exists. E-SINDy requires extensive hyperparameter tuning to achieve convergence on this dataset. Conversely, our proposed structural algorithm autonomously identifies the equations in equation 15 without any manual recalibration.

**ProbTS Competition Benchmarking.** To further validate the framework's capability on complex, real-world data, we benchmarked it using datasets from the ProbTS competition Zhang et al. (2024). We first detail the framework's behavior on the `Exchange Rate NIPS` dataset, visualized in Fig. 3.

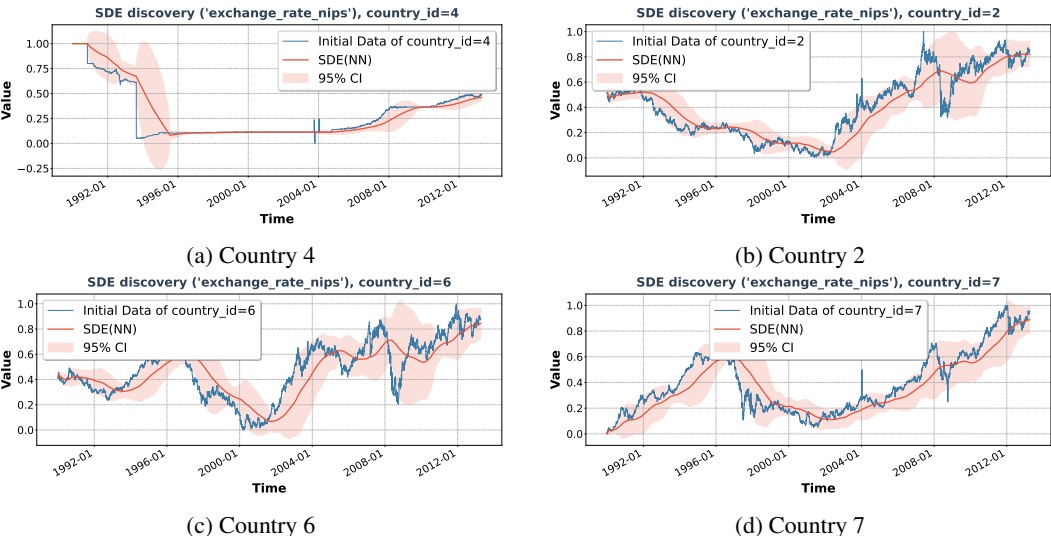

(a) Country 4        (b) Country 2

(c) Country 6        (d) Country 7

Figure 3: Structural SDE discovery results on the NIPS exchange rate dataset (blue lines) across distinct macroeconomic systems. The red line represents the expected mean trajectory, enveloped by a 95% confidence interval.

The time series for Country 4 exhibits a highly atypical exchange rate evolution. This abrupt dynamic shift triggers the algorithm to generate a piecewise structural model consisting of two distinct equations to correctly sample the solution trajectory, as formalized in equation 18. An ensemble of 50 equations was utilized for this reconstruction.

$$\begin{cases} A(t)\dot{x} + B(t) = \dot{x} \cdot \cos\left(C(t)t + D(t)\right), & t < T_1 \\ E(t) \cdot x + F(t) = \dot{x}, & t \geq T_2 \end{cases} \tag{18}$$

Here, $x = x(t)$ denotes the time series, with $\dot{x}$ and $\ddot{x}$ acting as the first and second temporal derivatives, respectively. The corresponding solution trajectories are visualized in Fig. 3a.

Unlike Country 4, Countries 2, 6, and 7 do not contain abrupt change points. Instead, they operate within three stable structural motifs, summarized in equation 19. These groups were used to evenly sample the equation space, with each specific structural form sampled and solved 40 times.

$$A(t)\dot{x}x + B(t)\dot{x} + C(t)x + D(t) = \ddot{x}\dot{x}$$
$$E(t)\dot{x} + F(t) = \dot{x}x \qquad (19)$$
$$G(t)x + H(t) = \dot{x}x$$

The structural similarities in equation 19 confirm that closely related time-series phenomenologies naturally yield mathematically analogous models. However, the internal parameter distributions differ significantly depending on the specific dynamics. For instance, Country 2 (Fig. 3b) exhibits distinct harmonic oscillations; this inherently activates a harmonic function representation, resulting in the dominant discovery of the second-order differential equation shown in equation 19. In this specific instance, the other candidate equations yield substantially lower mean coefficient weights. Conversely, Countries 6 and 7 (Fig. 3c and Fig. 3d) present vastly different coefficient distribution profiles and are therefore primarily governed by the first-order equations listed in equation 19.

A comprehensive statistical evaluation of our approach against leading ProbTS competition baselines Zhang et al. (2024) is detailed in Tab. 1 (optimal values highlighted in bold). Where applicable, we recorded Mean Squared Error (MSE) and Continuous Ranked Probability Score (CRPS). With the sole exception of our structural model, all baseline evaluations utilized an input/output `window_size` of 32. Differences in window sliding mechanics dictate two distinct modes of time-series restoration: Autoregressive (AR), where the subsequent window is predicted based on the model's prior predictions, and non-Autoregressive (non-AR), where predictions rely continuously on ground-truth lookbacks. For exact hyperparameter sets and technical realizations, we direct readers to the supplementary source code.

Specifically, the `NeuralODE` baseline utilizes the standard implementation from Chen et al. (2018). Because NeuralODE is a strictly deterministic architecture, only MSE was computed. Implementations for `NeuralSDE` and `gru` strictly followed the configurations described in Oh et al. (2024). For the NeuralSDE baseline, ten distinct architectures were evaluated, and Tab. 1 reports only the absolute best-performing configuration per dataset. Finally, the `deepar` baseline refers to the GluonTS `DeepAREstimator` model Alexandrov et al. (2020).

Table 1: Experimental validation of the approach and comparison with neuralODE and neuralSDE

| Dataset | Model | non-AR MSE | AR MSE | non-AR CRPS | AR CRPS | Model Size |
|---|---|---|---|---|---|---|
| | NeuralSDE (neuralsde_2_00) | 4.28e-03 | **0.0800** | 0.0476 | **0.2112** | 5362 |
| | NeuralODE | 0.0569 | 0.1766 | N/A | N/A | 4835 |
| electricity | Structural Model | 0.0252 | N/A | 0.0845 | N/A | 8 |
| | deepar | 0.0566 | 0.4163 | 0.1640 | 0.4513 | 25884 |
| | gru | **4.12e-03** | 0.5675 | **0.0472** | 0.5342 | 4801 |
| | NeuralSDE (neuralsde_0_00) | **2.28e-04** | 0.1408 | **9.54e-03** | 0.2905 | 4834 |
| | NeuralODE | 1.54e-03 | 0.6415 | N/A | N/A | 4835 |
| exchange | Structural Model | 5.05e-03 | N/A | 0.0329 | N/A | 8 |
| | deepar | 2.02e-03 | 0.4390 | 0.0256 | 0.5137 | 25884 |
| | gru | 4.72e-04 | **0.1346** | 0.0137 | **0.2836** | 4801 |
| | NeuralSDE (neuralsde_2_00) | 0.0128 | 0.5096 | 0.0770 | 0.4266 | 5362 |
| | NeuralODE | **5.62e-03** | 0.3758 | N/A | N/A | 4835 |
| solar | Structural Model | 0.0832 | N/A | 0.1637 | N/A | 8 |
| | deepar | 0.2155 | **0.1293** | 0.3289 | **0.2151** | 25884 |
| | gru | 0.0118 | 0.2273 | **0.0713** | 0.3351 | 4801 |
| | NeuralSDE (neuralsde_1_00) | **3.56e-03** | 0.3240 | **0.0366** | 0.3353 | 4834 |
| | NeuralODE | 4.16e-03 | **0.0425** | N/A | N/A | 4835 |
| traffic | Structural Model | 0.0116 | N/A | 0.0531 | N/A | 8 |
| | deepar | 0.0234 | 6.37e+09 | 0.1060 | 1.60e+04 | 25884 |
| | gru | 3.67e-03 | 0.1006 | 0.0375 | **0.2261** | 4801 |
| | NeuralSDE (neuralsde_0_00) | **2.49e-03** | 0.2411 | **0.0247** | 0.1566 | 4834 |
| | NeuralODE | 4.58e-03 | **7.74e-03** | N/A | N/A | 4835 |
| wiki2000 | Structural Model | 0.0045 | N/A | 0.0531 | N/A | 8 |
| | deepar | 5.37e-03 | 0.0224 | 0.0371 | 0.0956 | 25884 |
| | gru | 2.77e-03 | 0.0101 | 0.0263 | **0.0662** | 4801 |

As indicated in Tab. 1, heavily parameterized neural architectures can achieve superior localized metrics. However, our structural learning approach operates under radically different and exponentially more restrictive informational constraints. Most critically, our framework is provided only with two

initial scalar conditions—$x(0)$ and $\dot{x}(0)$—whereas the AR and non-AR evaluation modes for the neural baselines fundamentally require a sliding historical context window of size 32.

Furthermore, our structural framework exhibits extreme parameter efficiency, utilizing merely $\sim 8$ variables, which is several orders of magnitude smaller than the thousands of learned weights required by DeepAR, GRU, or NeuralSDE models. Despite this profound architectural simplicity, our model provides a highly competitive and fundamentally distinct paradigm for time-series analysis. On the `exchange` dataset, for instance, it performs within a single order of magnitude of the absolute best neural baseline. This resilience directly stems from its core philosophy: rather than interpolating data through millions of opaque weights, it distills the entire forecasting process into explicitly discovered governing differential equations equation 19. This advantage is particularly evident in the non-autoregressive (non-AR) forecasting mode—a rigorous test of a model's capacity to internalize the global, macroscopic dynamics of a series rather than merely extrapolating local trends. On complex datasets like `traffic` and `wiki2000`, our model's MSE and CRPS scores stand highly competitive against state-of-the-art neural networks. Ultimately, these results confirm that automatically discovered structural equations offer an exceptionally robust, transparent, and accurate representation of true system mechanics.

## 5    CONCLUSION

In summary, this paper presents a novel structural learning framework for the data-driven analysis of complex, inherently noisy dynamical systems. By recasting automated equation discovery as a probabilistic inference task, our approach successfully transitions from generating disjointed, deterministic equation ensembles (such as those produced by E-SINDy) to extracting unified, mathematically interpretable Stochastic Differential Equations.

The proposed methodology introduces a fundamental paradigm shift in time-series modeling. Standard machine learning architectures excel at local, short-term interpolation but inherently function as opaque black boxes, memorizing localized data correlations. In contrast, structurally learned SDEs act as global models. They encapsulate the explicit governing laws and internal mechanics of the entire macroscopic process. By isolating these core physical structures, our framework achieves highly competitive baseline precision (MSE and CRPS) despite operating under severe informational constraints and utilizing orders of magnitude fewer parameters than contemporary neural networks.

We acknowledge an inherent computational trade-off in this approach: the evolutionary synthesis, statistical aggregation, and structural filtering required to discover these SDEs demand a higher initial acquisition and training time compared to standard gradient-based parameter optimization. However, this computational investment yields a profound advantage. The resulting compact SDE models are vastly superior at capturing the authentic stochastic nature of real-world time series without sacrificing transparency.

Ultimately, by providing a robust and automated pathway to discover compact, highly interpretable, and globally valid SDE models directly from raw data, this structural learning framework unlocks new frontiers for rigorous scientific reasoning. Furthermore, it explicitly bridges the gap between purely data-driven forecasting and advanced automated symbolic manipulation systems, enabling a deeper mathematical understanding of complex phenomena.

## ACKNOWLEDGEMENTS

The research was carried out within the state assignment of Ministry of Science and Higher Education of the Russian Federation (project FSER-2024-0004).

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

## A  DETAILED SCHEME OF APPROACH

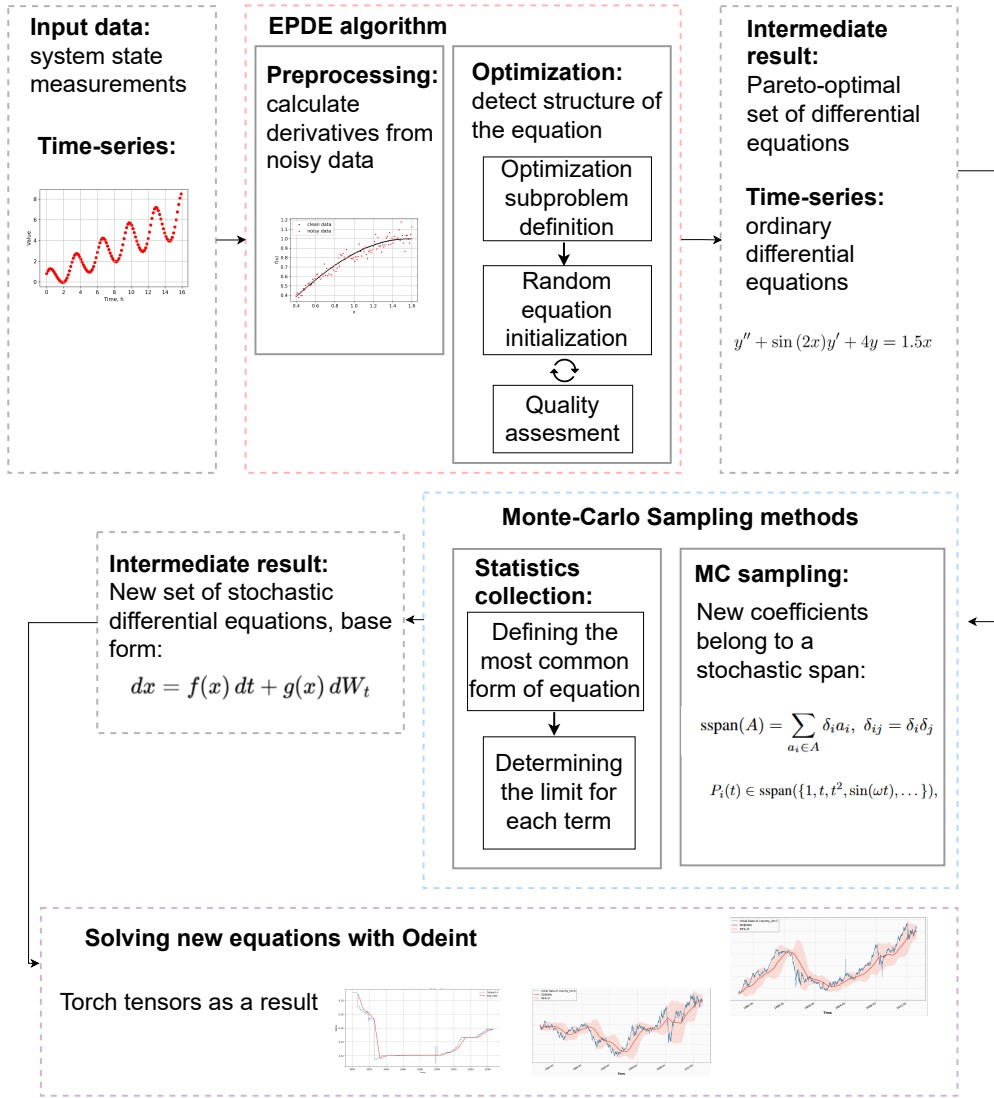

Figure A.1: Detailed algorithm scheme

