# OpenReview forum: "Automated structural learning of stochastic dynamics"
_mathai.club/MathAI/2026/Conference — 2026 Oral_

### Official Review · Reviewer_kUsT · 2026-03-11
**Fundamental mathematical errors and flawed validation undermine the entire framework**

**Rating:** 2
**Confidence:** 4

**Review:**

Summary
This paper proposes a framework that recasts equation discovery as a probabilistic inference task, mapping structural variability across multiple symbolic regression (EPDE) executions into continuous probability distributions to construct Stochastic Differential Equations (SDEs). While the motivation—bridging deterministic symbolic regression and stochastic modeling—is interesting, the paper suffers from critical mathematical errors in its foundational definitions, misapplication of core theorems, and a fundamentally flawed validation methodology.
Strengths

The core idea of leveraging ensemble variability from multiple symbolic regression runs to construct SDEs is novel and potentially impactful.
The paper addresses a genuine gap between deterministic equation discovery and stochastic systems.
The application to real-world exchange rate data demonstrates ambition beyond toy examples.

Weaknesses
Critical Mathematical Errors

Equation (8) — Malformed stochastic span definition. The product notation δij=δiδj\delta_{ij} = \delta_i \delta_j
δij​=δi​δj​ is undefined. Is this element-wise, Kronecker, or Hadamard product? This is a foundational definition upon which the entire framework rests, yet it is never clarified.

Equation (9) — Circular definition. σi(t)\sigma_i(t)
σi​(t) appears on both sides: δi=μi(t)+σi(t)\delta_i = \mu_i(t) + \sigma_i(t)
δi​=μi​(t)+σi​(t) where σi(t)∼N(0,σi(t))\sigma_i(t) \sim \mathcal{N}(0, \sigma_i(t))
σi​(t)∼N(0,σi​(t)). A random variable cannot be drawn from a distribution parameterized by itself. This violates basic probability theory.

Equation (11) — Unjustified jump to Wiener processes. The transition from KK
K deterministic ODE solutions (Eq. 6) to continuous Wiener processes dWidW_i
dWi​ is completely unjustified. How are dWidW_i
dWi​ extracted from a discrete ODE ensemble? No answer is provided.

Central Limit Theorem misapplication. The paper invokes the CLT to justify normality of coefficient distributions. However, the ensemble comes from running a deterministic algorithm (EPDE) multiple times—these are algorithm outputs, not i.i.d. samples from a fixed distribution. CLT conditions are never verified.
Wong-Zakai theorem misapplication. The theorem concerns smoothing approximations of Brownian motion and does NOT authorize using arbitrary ODE ensembles to approximate SDEs. Specific regularity conditions are required but never verified.
Confidence interval notation error. The paper writes μ±1.96σ2\mu \pm 1.96\sigma^2
μ±1.96σ2 for 95% CI. The correct formula is μ±1.96σ\mu \pm 1.96\sigma
μ±1.96σ. Using σ2\sigma^2
σ2 has wrong units and wrong magnitude.


Validation Methodology Failures

Incomparable evaluation in Table 1. The structural model uses only 2 initial conditions [x(0),x˙(0)][x(0), \dot{x}(0)]
[x(0),x˙(0)] while neural baselines use a sliding window of size 32. These are fundamentally different evaluation protocols presented as equal competition.

Ground truth from lecture notes. Equation (17) benchmarks against Howard (2009), which is undergraduate lecture notes (Math 442), not peer-reviewed research. Using lecture notes as validation ground truth is academically inappropriate.
Category error in benchmark selection. ProbTS datasets (exchange rates, electricity, traffic) are time-series forecasting benchmarks, not naturally SDEs. Using them to validate SDE discovery is a category error.
Abstract contradicted by results. The abstract claims a "single, cohesive SDE" but results show piecewise models (Eq. 18) and per-country equations (Eq. 19)—multiple equations, not a unified structure.

Missing Elements

No proofs for existence, uniqueness, or convergence claims. The commutative diagram (Eq. 12) contains question marks.
No comparison to established SDE discovery methods (drift-diffusion estimation, kernel-based SDE discovery, Bayesian SDE inference). Only compared to E-SINDy, which is designed for ODE discovery.
No ablation study on ensemble size KK
K (claimed "optimal" at K=50K=50
K=50 without evidence).

No synthetic experiments with known SDE ground truth.

LLM-Assisted Writing Concerns
Moderate probability (60–70%) of LLM-assisted writing. Excessive superlatives ("paradigm shift," "seamlessly bridges," "profound analytical power," "unlock new frontiers") combined with lack of standard SDE terminology (missing: Itô integrals, quadratic variation, martingale properties, adapted processes) suggest text generation assistance.
Reference Issues

Suryadevara & Yanamala (2020) cites "Revista de Inteligencia Artificial en Medicina"—this journal name is suspicious and could not be independently verified.
Howard (2009) is lecture notes, inappropriate as ground truth source.

Questions for Authors

Please provide a rigorous definition of the product notation in Eq. (8).
How do you resolve the circularity in Eq. (9)?
What formal justification connects the ODE ensemble to Wiener processes in Eq. (11)?
Why was σ2\sigma^2
σ2 used instead of σ\sigma
σ in the confidence interval formula?

Can you provide convergence proofs with explicit rates?

Limitations
The authors do not adequately address limitations. The piecewise model contradiction, lack of theoretical guarantees, and benchmark mismatch are not discussed.
Ethics
No ethical concerns identified.
Overall Assessment
The mathematical foundations of this paper are unsalvageable in their current form. Seven critical errors in core equations, misapplication of two fundamental theorems (CLT and Wong-Zakai), and a validation methodology built on incomparable evaluation protocols and inappropriate benchmarks prevent acceptance. The paper requires a complete theoretical reconstruction.

---

> ### Author Rebuttal · Authors · 2026-03-13
>
> We sincerely thank the reviewer for their detailed feedback. We agree the paper's theoretical foundations could be more strictly formalized. Comments regarding mathematical rigor are entirely valid within classical SDE theory, where models are typically variations of a standard random walk. However, our primary objective is providing a practical ML tool that generates symbolic models as SDEs and random differential equations.
> This significantly expands the discoverable model variety, differentiating modern ML from classical inverse problems. Strictly formalizing this extended space would require a separate theoretical paper. Thus, we acknowledge certain theoretical steps are somewhat "hand-waved" to prioritize computational and practical utility.
> Regarding the Wong-Zakai theorem critique: a strictly classical application would inherently constrain us to standard SDEs with random walks. Our adaptation is a deliberate effort toward a generalized case, preserving required structural flexibility. We will explicitly clarify this choice and its theoretical trade-offs in the revision.
>
> Finally, while the review focused on theoretical strictness, we wish to highlight overlooked empirical contributions:
> 1. Parameter efficiency: unlike predictive methods like ProbTS, our algorithm excels in interpolation, achieving state-of-the-art neural network accuracy with drastically fewer parameters.
> 2. Data efficiency: unlike neural networks requiring sliding windows and extensive history, our approach models dynamics using only initial conditions.
>
> We hope the reviewer appreciates this work as a practical ML tool for equation discovery. We will gladly incorporate their theoretical caveats to clearly define our mathematical boundaries.

---

### Official Review · Reviewer_UQik · 2026-03-12
**Automated Structural Learning of Stochastic Dynamics**

**Rating:** 7
**Confidence:** 3

**Review:**

Summary

The reviewed paper addresses a fundamental problem in data-driven dynamical systems modeling. The proposed solution is the automated discovery of interpretable stochastic differential equations (SDEs) directly from noisy time series data. A detailed analysis of the current state of the problem is provided. The authors demonstrate a critical methodological gap between deterministic equation discovery (e.g., SINDy, EPDE), which ignores the inherent stochasticity of systems, and opaque neural network approximations (e.g., NeuralSDEs), which sacrifice mathematical transparency. Ensemble methods like E-SINDy, while capturing parametric uncertainty, generate unwieldy collections of ODEs unsuitable for formal analysis.
To solve this problem, the authors propose a novel framework that reframes equation discovery as a probabilistic inference task. The novel approach lies in exploiting the intrinsic stochasticity of evolutionary optimization. It maps the structural variability observed across multiple runs of symbolic regression onto continuous probability distributions over mathematical terms and coefficients. This allows for the aggregation of a heterogeneous ensemble of deterministic ODEs into a single, compact, and fully interpretable SDE. The framework is validated on a stochastic Lotka-Volterra model and benchmarked against state-of-the-art methods (NeuralODE, NeuralSDE, DeepAR) using the ProbTS datasets. The results demonstrate that the proposed structural model achieves competitive performance in probabilistic forecasting (CRPS) while maintaining complete mathematical transparency and remarkable data efficiency.

Strengths

The paper presents a conceptually original solution. Ensemble variability is treated not as computational noise, but as information about the system's intrinsic stochasticity. The work successfully combines sparse identification of dynamics and probabilistic modeling.
The experimental results convincingly demonstrate that the structural model captures the underlying dynamical law rather than temporal correlations. Using specific examples, the paper shows the capabilities of the proposed framework and its advantages over neural network baselines.
An empirical evaluation of the framework's performance was conducted by comparing it against several baseline models (NeuralODE, NeuralSDE variants, GRU, DeepAR) on standard datasets from a recognized competition (ProbTS) using appropriate metrics (MSE, CRPS). This confirms the practical utility of the method.

Weaknesses

The quality of the final SDE is fundamentally contingent on the exploratory power of the base symbolic regression algorithm. If the initial ensemble, due to poor hyperparameters or algorithmic limitations, fails to explore relevant regions of the structural space, the inferred probability distributions will be incomplete. This is acknowledged in the paper's discussion and requires further investigation.
The description of the transition from empirical histograms to continuous distributions lacks sufficient detail. Furthermore, it is unclear whether the coefficients of different terms are assumed to be independent. A more detailed explanation of the functional form of the distribution would be desirable.

Recommendation

This manuscript and the proposed framework represent an interesting, well-motivated, and empirically validated approach in the field of data-driven dynamical systems modeling. The strengths, particularly in terms of interpretability and data efficiency, warrant recommending the paper for publication.
To strengthen the article before final publication, the authors are recommended to add a brief comparison of the training/inference time of the structural model versus the neural network baselines.

---

### Official Review · Reviewer_M6Qg · 2026-03-13
**Novel SDE discovery framework with strong empirical results but incomplete mathematical rigor**

**Rating:** 7
**Confidence:** 3

**Review:**

This paper proposes automated SDE discovery by aggregating symbolic regression ensembles (EPDE/SINDy) into probabilistic coefficient distributions. The method produces compact, interpretable SDEs validated on Lotka-Volterra and ProbTS benchmarks, achieving competitive accuracy with small number of parameters versus thousands in neural baselines.

**Strengths:**
-  Treating ensemble variability as informative signal rather than noise is creative and addresses a real gap between deterministic symbolic regression and stochastic modeling
- Strong empirical validation with ProbTS datasets plus Lotka-Volterra, and competitive MSE/CRPS despite extreme parameter efficiency
- Outputs based on human-readable equations (Eq. 15, 18, 19) unlike Neural SDEs
- Improved data efficiency by using only $[x(0), \dot{x}(0)]$
versus sliding windows (size 32) for neural baselines
- Honestly acknowledges computational costs and trade-offs

**Weaknesses:**
- Circular definition (Eq. 9) $\sigma_i(t) \sim \mathcal{N}(0, \sigma_i(t))$
 violates probability theory (likely meant $\sigma_i(t) \cdot \varepsilon$ where $\varepsilon \sim \mathcal{N}(0,1)$)
- Confidence intervals (Eq. 15) uses μ ± 1.96σ² instead of μ ± 1.96σ (wrong units)
- In regards to clustering logic, structural modes mentioned but never formalized—no algorithm, no convergence guarantees
- Section 2 claims "Existence, Uniqueness, and Convergence" but provides no proofs; commutative diagram (12) contains literal question marks
- "Threshold-based filtering" undefined (regarding Hyperparameter opacity)

**Assesment:**
The novel approach, strong empirical results, and extreme parameter efficiency merit publication, but theoretical gaps must be addressed. Practical utility outweighs incomplete rigor for an applied mathematics venue, as long as framing of limitations is honest. The quality of paper can be swiftly improved with revision by fixing theoretical gaps (e.g. Eq. 9), defining applicable scope (engineering tool vs mathematical framework), and clarifying convergence claims and related open problems.

---

### Decision · Program_Chairs · 2026-03-14

**Decision:**

Accept (Oral)

**Comment:**

Dear Author(s),

On behalf of the Program Committee of the International Conference on Mathematics of Artificial Intelligence (MathAI 2026), we are pleased to inform you that your paper has been accepted for an oral presentation at MathAI 2026.

Your paper was evaluated through a rigorous two-stage review process involving both automated screening and expert review by members of the Program Committee. The reviewers recognized the quality and contribution of your work.

Presentation details:

- Format: Oral presentation (15–20 minutes + 5 minutes Q&A)
- Mode: You may present either in person (offline) at the conference venue in Sirius, Russia, or remotely via Zoom. Please indicate your preferred mode when confirming your participation.
- Conference dates: Marh 30 - April 3, 2026
- Website: https://mathai.club

Next steps:

1. Please confirm your participation and presentation mode by replying to this email mathai.club@yandex.ru no later than March 15, 2026 18:00 Moscow time.
2. If you plan to attend in person, the organizing committee will provide accommodation details separately.
3. Please prepare your final camera-ready manuscript according to the formatting guidelines available at https://mathai.club and upload it to OpenReview by March 15, 2026 18:00 Moscow time.

Should you have any questions regarding the program, logistics, or your presentation slot, please do not hesitate to contact us.

We look forward to your contribution to MathAI 2026.

With kind regards,

MathAI 2026 Program Committee
International Conference on Mathematics of Artificial Intelligence
https://mathai.club
OpenReview: https://openreview.net/group?id=mathai.club/MathAI/2026/Conference
Telegram: https://t.me/MathAI_club
Email: mathai.club@yandex.ru